# Communications between Neutrophil–Endothelial Interaction in Immune Defense against Bacterial Infection

**DOI:** 10.3390/biology13060374

**Published:** 2024-05-24

**Authors:** Zhigang Sun, Ruoyi Lv, Yanxin Zhao, Ziwen Cai, Xiaohui Si, Qian Zhang, Xiaoye Liu

**Affiliations:** 1Beijing Key Laboratory of Traditional Chinese Veterinary Medicine, Beijing University of Agriculture, No. 7 Beinong Road, Changping, Beijing 102206, China; sunzhigang@bua.edu.cn (Z.S.); zhaoyanxin@bua.edu.cn (Y.Z.); caiziwen@bua.edu.cn (Z.C.); xiaohuisi@bua.edu.cn (X.S.); 2Animal Science and Technology College, Beijing University of Agriculture, No. 7 Beinong Road, Changping, Beijing 102206, China; lvruoyi@bua.edu.cn; 3Beijing Traditional Chinese Veterinary Engineering Center, Beijing University of Agriculture, No. 7 Beinong Road, Changping, Beijing 102206, China

**Keywords:** neutrophil–endothelial interaction, bacterial infection, neutrophil clearance, endothelial activation, endothelial dysfunction

## Abstract

**Simple Summary:**

Immune defense against bacterial infection involves a fierce struggle between bacteria and immune cells. Neutrophils, as the first line of defense, must cross the epithelial barrier to reach the infected site. During this process, communication between neutrophils and endothelial cells is crucial for regulating immune defense, cascade reactions, and inflammation. Therefore, we present the detailed steps of neutrophil–endothelial cell interactions and the principles of neutrophil biology. The overview of the research progress focuses on transendothelial neutrophil killing and how endothelial dysfunction affects inflammatory cascade reactions. These summaries of neutrophil activation and endothelial dysfunction provide potential therapeutic targets for discovering and developing antibacterial and anti-inflammatory drugs.

**Abstract:**

The endothelial barrier plays a critical role in immune defense against bacterial infection. Efficient interactions between neutrophils and endothelial cells facilitate the activation of both cell types. However, neutrophil activation can have dual effects, promoting bacterial clearance on one hand while triggering inflammation on the other. In this review, we provide a detailed overview of the cellular defense progression when neutrophils encounter bacteria, focusing specifically on neutrophil–endothelial interactions and endothelial activation or dysfunction. By elucidating the underlying mechanisms of inflammatory pathways, potential therapeutic targets for inflammation caused by endothelial dysfunction may be identified. Overall, our comprehensive understanding of neutrophil–endothelial interactions in modulating innate immunity provides deeper insights into therapeutic strategies for infectious diseases and further promotes the development of antibacterial and anti-inflammatory drugs.

## 1. Introduction

Upon encountering infection, hosts undergo a drastic struggle between pathogenic bacteria and immune cells [1,2]. Innate immunity is the main weapon used by the host defense to combat the invasion of pathogenic bacteria [3,4]. The endothelium, as an abundant organ, consists of a barrier that closely cooperates with immune cells to combat bacterial infection. Therefore, cell-to-cell communications play a key role in triggering cross-linking signaling pathways [5]. Neutrophil–endothelial interactions are not only essential for immune cell arrival at infected sites, but also influence the efficacy of bacterial clearance and the release of pro-inflammatory chemokines [6,7,8]. Currently, a lot of reports show that the maintenance of endothelial barrier integrity plays a key role in neutrophil and endothelial activation [9,10,11,12,13]. In contrast, damage to endothelial integrity will lead to endothelial dysfunction and further impair the activation of transendothelial neutrophils [2,5,14]. Furthermore, the excessive activation of neutrophils and the formation of neutrophil extracellular traps (NETs) also cause endothelial dysfunction [15,16]. 

Indeed, the endothelium performs various roles in the neutrophil clearance of bacterial infection during neutrophil–endothelial cell interactions [7,8,16]. The endothelial barrier forms a defense line of endothelial cells to protect the host and summon immune cells like neutrophils in innate immunity. Undoubtedly, endothelial cells are essential for the activation of neutrophils in the defense against bacterial infection [17,18]. Mechanism research is increasingly being carried out on the interactions between endothelial cells and neutrophils [5,18,19]. Researchers urgently want to clarify the scientific issue behind how bacteria hijack through the defense of the innate immune system and why tissue injury or inflammation occurs. However, the impacts of endothelial dysfunction and activation on neutrophil defense and excessive inflammation remain unclear and require a system summary and in-depth discussion.

In this review, we demonstrate the essential steps of neutrophil combat against bacterial infection, including rolling, clawing, recruitment, and migration across endothelial cells. Additionally, we propose how neutrophil–endothelial cell interaction impacts both neutrophil defense and endothelial activation/dysfunction during bacterial hijacking. Finally, we provide an overview of the mechanisms underlying endothelial dysfunction and inflammatory cascades. Focusing on these specific mechanisms will aid in screening and identifying potential host targets for preclinical research on bacterial infection or the clinical discovery of antibacterial drugs.

## 2. Cooperative Behaviors of Neutrophil and Endothelial Cell

### 2.1. Neutrophil Generation and Circulating

Neutrophils are generated in bone marrow and derived from myeloid stem cells. As shown in Figure 1A, myeloid stem cells become myeloblasts, and then approximately 50–80% of myeloblast cells are divided into neutrophil progenitors and further formed neutrophil granulocytes [20,21]. In fact, neutrophils are the most abundantly generated cells among the innate immune cells. Activated neutrophils display increased lifespans through multitudinous survival factors, such as cytokines, chemokines, hormones, growth factors, lipid mediators, adhesion molecules, environment factors, pathogen-associated molecular patterns (PAMPs) and damage-associated molecular patterns (DAMPs) [14]. These survival factors ensure that activated neutrophils circulate in blood for a few hours to fulfill the defensive mission.

### 2.2. Neutrophil Rolling and Adhesion on Endothelium

Endothelial cells form the inner barrier layer that covers various organs, especially the brain, blood and lymphatic vessels [22]. The endothelium layer not only plays a role as a physical barrier, but also acts as a functional tissue to induce the activation of immune cells or the regulation of inflammation [17,18]. When bacteria enter into a host, immune cells such as neutrophils must cross endothelial cells to arrive at the site of bacterial entry. The further steps of neutrophil migration across the endothelial layer are tightly regulated processes [23,24]. In fact, mature neutrophils from bone marrow maintain the status of free circulating in vascular [20]. When the tissue is invaded by bacteria, endothelial cells and neutrophils are awakened and a series of behaviors occur between them. The initial step of neutrophil defense begins with neutrophil rolling on the endothelial cells, which sense the bacterial signals as engaging with the responding selectins and chemokines on the activated endothelial cells (Figure 1B(i)). For instance, the P-selectin and E-selectin expressed on endothelial cells sense and capture rolling neutrophils by binding the proteins of P-selectin ligand 1 (PSGL1), Pentraxin 3 (PTX3), E-selectin ligand 1 (ESL1), CD44 and L-selectin on neutrophils (Table 1) [20,23,25]. Furthermore, endothelial intercellular adhesion molecule-1 (ICAM-1) interacts with neutrophil β_2_-integrins to decelerate the rolling neutrophil. In addition, recent research shows that the endothelial-cell-derived CD95 ligand serves as a chemokine the in induction of neutrophil slow rolling and adhesion [6]. Subsequently, the β_2_-intergin on neutrophils such as LAF-1, Mac-1 and VLA-4 interacts with P-selectin and E-selectin or ICAM from endothelial cells to mediate neutrophil adhesion (Table 1 and Figure 1B(ii)) [7,20,26,27,28]. To enhance P-selectin-dependent neutrophil adhesion, neutrophils trigger endothelial signaling by releasing oncostatin M (OSMR) to bind with OSM receptors [7]. Moreover, reports have shown that endothelial or monocyte interleukin 1α (IL-1α) can also promote neutrophil adhesion and activation (Table 1) [29,30]. Lastly, neutrophil crawling and recruitment are performed on endothelial cells (Figure 1B(iii)) to gain firm adhesion and enable further migration across the endothelial barrier to arriveat the infected site [23,25]. For instance, recent research shows that G-protein coupled receptor (GPR35) facilitates neutrophil recruitment in response to serotonin metabolite 5-hydroxyindoleacetic acid (5-HIAA) [12].

### 2.3. Neutrophil Transendothelial Migration Strategies 

Upon neutrophil recruitment and the formation of firm adhesion, transendothelial migration (Figure 1B(iv)) depends on two major mechanisms that involve the penetration of individual endothelial cells and paracellular squeezing between neighboring endothelial cells [18,20]. Paracellular or transcellular migration are also subject to the interaction between the neutrophil β2 integrins LFA-1 and Mac-1 and endothelial molecules such as ICAM-1 or ICAM-2 (Table 1) [33]. Additionally, these adhesion molecules, including platelet endothelial cell adhesion molecule 1 (PECAM-1), junctional adhesion molecule (JAM) and endothelial cell-selective adhesion molecule (ESAM), on endothelial cells are also distributed on the surface of neutrophils, whose adhesion molecules all contribute to the neutrophil migration between endothelial cells (Table 1) [11,38]. Furthermore, the endothelial cell-expressed transient receptor potential melastatin-2 (TRPM2) activates Ca^2+^ signaling and VE-cadherin phosphorylation to mediate the transendothelial migration of neutrophils [18]. Notably, increasing evidence shows that endothelial cells are key regulators of neutrophil migration [8,12,13,34]. Nevertheless, endothelial activation/dysfunction depends on whether neutrophils are excessive activated [14,21,27]. In fact, activated neutrophils promise transendothelial neutrophils to combat bacteria though various means involving phagocytosis, degranulation and the formation of neutrophil extracellular traps (NETs) (as described in Figure 1B); these defense mechanism prolong the neutrophil lifespan and enable persistent interaction with endothelial cells, causing endothelial dysfunction.

## 3. Defense Bacterial Infection of Transendothelial Neutrophils

### 3.1. Clearance Strategies by Transendothelial Neutrophils

Neutrophils sense the signals of chemokines, cytokines and granular proteins derived from damaged endothelial cells to activate themselves (CD11b or CD11a positive) and further fulfill the mission of clearance (Figure 2). Due to the relatively short life of neutrophils, abundant cellular members need to immediately respond to the infection. Transendothelial neutrophils bear the mission and carry out a bactericidal function by performing phagocytosis, degranulation or using neutrophil extracellular traps (NETs), and so on (Figure 2). Intracellular and extracellular killing are the main functions of these short-lived neutrophils and a constant supply of transendothelial neutrophils must routinely be turned over. Therefore, to preserve the neutrophil homeostatic balance, neutrophil death is also an inevitable process in this battle between immune cells and bacteria. Meanwhile, no matter which ends of the neutrophil meet, diverse pathways are required to accelerate neutrophil intracellular and extracellular clearance [39,40].

### 3.2. Intracellular Clearance by Neutrophil Neutralization

After transendothelial migration, intracellular clearance by neutrophil neutralization is mainly relies on phagocytosis and efferocytosis [15,21,23]. Neutrophils encapsulate cytosolic bacteria in phagosomes and kill these pathogenic bacteria in two ways including the NADPH oxygenase-dependent pathway triggered by reactive oxygen species and antibacterial proteins that kill bacteria, such as cathepsin, lysozyme, defensin and lactoferrin [20,21,27,41]. Moreover, during phagocytosis, neutrophils degranulate and release a series of substances that kill bacteria and cause damage to the blood vessels and surrounding tissues. These particles include primary particles (azurophilic particles), containing myeloperoxidase, defensins, a variety of serine proteases, etc.; secondary particles (special particles) containing lysozyme, lactoferrin (inhibition of iron-requiring bacteria), neutrophil collagenase, OLFM4, etc.; tertiary granules, containing lysozyme, type IV collagenase, etc. [20,21,24,35].

In addition, the other major intracellular clearance strategy is neutrophil autophagy (Figure 2, blue cycle line), which utilizes the autophagosome to uptake invaded bacteria and further degrade bacteria by lysosomes [42,43]. In particular, neutrophil autophagy is not only responsible for bacterial clearance, but also contributes to maintaining homeostasis by eliminating denatured proteins or damaged organelles [44]. Bacteria entering the phagolysosome can be killed by oxygen-dependent mechanisms (reactive oxygen species ROS and reactive nitrogen RNS) and oxygen-independent mechanisms (lactoferrin, lysozyme, α-defensin, etc.), and finally hydrolyzed by acid hydrolases. The mitochondria of human neutrophils, along with NADPH oxidase, act as bactericidal ROS generators that induce the killing of *Staphylococcus aureus* [45].

### 3.3. Extracellular Clearance by Neutrophil Extracellular Traps

The formation of NETs is vital for the extracellular clearance of transendothelial neutrophil. These events involve the expulsion of the intracellular contents inside neutrophils extracellularly for the entrapment and eventual killing of bacteria (Figure 2, green cycle line). These intracellular contents are assembled on the scaffold of decondensed chromatin [26]. Precisely, NET is composed of nuclear chromatin, citrullinated histones, elevated concentrations of highly potent bactericidal azurophilic and specific granular proteins, namely proteolytic enzymes that degrade bacterial virulence factors, as well as enzymatically active myeloperoxidase [46]. In addition, the majority of DNA in NETs originates from the nucleus and mitochondrial DNA [47]. It worth noting that NETs are released not only for the trapping and killing of bacteria, but that they also can induce endothelial cell activation via the production of interleukin 1α (IL-1α) and cathepsin G [48]. NETs promote interleukin 8 (IL-8) release for the recruitment of other immune cells, such as M1-like macrophage [49]. However, research indicates that NETs amplify and propagate endothelial dysfunction [29,30,50]. On the other hand, prolonged infection and the death of neutrophils can also cause endothelial dysfunction and lead to inflammation.

### 3.4. Neutrophil Death

Under physiological conditions, neutrophils circulate in the blood or tissues and have a relatively short lifespan of approximately 18–19 h. Upon bacterial invasion, neutrophil death occurs after combating pathogenic bacteria. Furthermore, neutrophil lysis following defense-mediated cell death involves necroptosis, proptosis, apoptosis, and NETosis to disrupt homeostatic balance (Figure 2, red cycle dotted line). The proper removal of deceased cells is crucial for avoiding unintended inflammatory injury [25]; otherwise, dead neutrophils further accelerate endothelial cell dysfunction and cause excessive inflammation. These complex death processes are very precisely regulated by the proteins expressed on both neutrophil and endothelial cells [7,51,52,53]. Finally, these endothelial cells eventually go in two different kinds of directions, becoming involved in endothelial activation and dysfunction [13,54,55]. Of these, activated endothelial cells promote neutrophil clearance and guarantee noninflammatory occurrence [30]. In contrast, endothelial cell dysfunction can induce proinflammatory responses through the activation of glycolysis, NADPH production, ROS or NO release, NLRP3 inflammasome formation, etc. [14,16,17,54]. Uncontrollable bacterial infection is a consequence of inflammation, which is regulated through cellular communications. Therefore, addressing inflammation necessitates the resolution of the pathway cascades involved in endothelial dysfunction.

## 4. Inflammatory Cascades in Endothelial Dysfunction and Potential Targets

### 4.1. Inflammatory Cascades

Neutrophils, serving as the primary effector cells of the innate immune response, are initially responsible for bacterial clearance and host cell protection. However, upon activation, neutrophils continuously release inflammatory mediators and disrupt neutrophil homeostasis. Furthermore, the formation of neutrophil extracellular traps (NETs) by dying neutrophils also contributes to endothelial dysfunction and subsequent inflammatory injury. Therefore, tightly regulated inflammation cascades play a crucial role in preventing unintended inflammatory damage. As shown in Figure 3, both pathogen-associated molecular patterns (PAMPs) and damage-associated molecular patterns (DAMPs) can trigger inflammatory signaling pathways. High mobility group box-1 protein (HMGB1) and inflammatory cytokines (tumor necrosis factor (TNF)-α), which belong to DAMPs, induce nuclear factor-κB (NF-κB) signaling (Figure 3i,ii) [56,57,58,59]. PAMPs such as bacterial LPS initiate TLRs (TLR4), not only activating the NF-κB pathway, but also driving endothelial cells toward reprograming the pro-inflammatory phenotype. In addition, platelet TLR4 activates NETs to ensnare bacteria in septic blood [60]. PAMPs also encompass other bacterial virulence factors, such as pore-forming toxins (PFTs), which use intracellular ROS to perpetuate endothelial cell injury or activation [24,27,61].

The production of ROS may damage mitochondria and induce the activation of the NLRP3 inflammasome complex [62,63,64,65]. Specifically, caspase-1, as a downstream protein, is activated by NLRP3 and further induces the gasdermin D-dependent pyroptosis pathway [66,67,68]. NETs promote MPO/H_2_O_2_ activation and induce cellular pyroptosis (Figure 3iii) [69,70] Moreover, NETs degrade the endothelial glycocalyx, increasing permeability and causing the collapse of the endothelial barrier. This ultimately leads to inflammatory dysregulation, resulting in insufficient microcirculatory blood flow, inadequate tissue perfusion, and life-threatening organ failure during the later stages of sepsis [71,72,73]. On the other hand, the main way to promote inflammation occurrence is through pyroptosis, which can be induced by the accumulation of intracellular LPS triggered by non-canonical Gram-negative bacteria. This activation leads to the cleavage of Gasdermin D into the pore, resulting in pyroptosis that enhances the endothelial cell inflammatory response (Figure 3iv) [67,74]

Altogether, the regulation of inflammatory cascades is primarily mediated by the interactions between neutrophils and endothelial cells. Maintaining neutrophil homeostasis and preventing endothelial dysfunction are key strategies for anti-inflammatory interventions.

### 4.2. Potential Targets

To summarize, determining the mechanisms behind inflammatory cascades would aid in identifying potential targets for anti-inflammatory drugs. It is crucial to explore breakthroughs in the interaction between endothelial cells and neutrophils. The first step in the prevention of inflammation is directly reducing neutrophil adhesion or recruitment to the endothelium. β2-integrins become potential targets due to their ability to perform neutrophil adhesion and recruitment (Figure 1 and Table 2). The β2-integrin inhibitors including soluble uric acid, hyperbaric oxygen, angiostatin and herbimycin A, can directly interact with neutrophil β2-integrins to hinder neutrophil recruitment [75,76,77,78] (Table 2). On the other hand, NETs are the key factors required for inflammatory occurrence (Figure 3). Therefore, the suppression of the formation of NETs is a potential target for anti-inflammatory strategies. For instance, PAD4 inhibitors such as Cl-amidine or GSK484 can inhibit NET formation to decrease the NET-mediated inflammation [79,80,81,82]. The phosphodiesterase inhibition of 6-gingerol shows the NET inhibition as well [83]. Reports show that nNIF and NRPs reduce inflammation by blocking NET formation in vitro and in mouse models of infection, and blocking systemic inflammation [84]. It is noteworthy that other inhibitors of NET formation, such as DNase1 and DNase1-like 3, promote anti-inflammatory effects by preferentially cleaving proteins to release DNA and degrade the scaffold of NET structures [85]. Finally, most investigations of anti-inflammatory means attempt to maintain ROS balance. During sepsis, the accumulation of ROS constitutes a significant mechanism for endothelial cell dysfunction (Figure 3). The antioxidant agents can serve as potential targets for anti-inflammation. For instance, GSH, NAC and Vitamin C scavenge ROS to reduce inflammation in both non-clinical and clinical trials [86,87,88]. Overall, these potential therapeutic targets could provide new opportunities for improved anti-inflammation medication.

## 5. Conclusions

Overall, we meticulously illustrate the crucial processes involved in the neutrophil’s battle against bacterial infection. These processes include the initial ‘rolling’ phase, where neutrophils adhere to the endothelium, followed by the ‘clawing’ stage, where the neutrophils extend their pseudopodia to firmly grasp the bacterial cells. Subsequently, we delve into the ‘recruitment’ phase, where the neutrophils are attracted to the site of infection by chemokines, and finally, the ‘migration’ stage, where they transmigrate across the endothelial cells to reach the bacteria.

Furthermore, we summarize the intricate relationship between neutrophils and endothelial cells, highlighting how their interaction influences both the neutrophil’s defensive capabilities and the endothelial cell’s activation/dysfunction during the bacterial invasion. This understanding of their interdependency paves the way for novel therapeutic strategies targeting both neutrophils and endothelial cells.

In the concluding part of this review, we provide an in-depth overview of the underlying mechanisms that lead to endothelial dysfunction and the subsequent inflammatory cascades. By focusing on these specific mechanisms, we aim to facilitate the screening and identification of potential host targets for preclinical research on bacterial infection, and potentially, the discovery of novel antibacterial drugs. This comprehensive approach towards unraveling the complex interplay between neutrophils and endothelial cells during bacterial infection offers valuable insights for future research and clinical applications.

## Figures and Tables

**Figure 1 biology-13-00374-f001:**
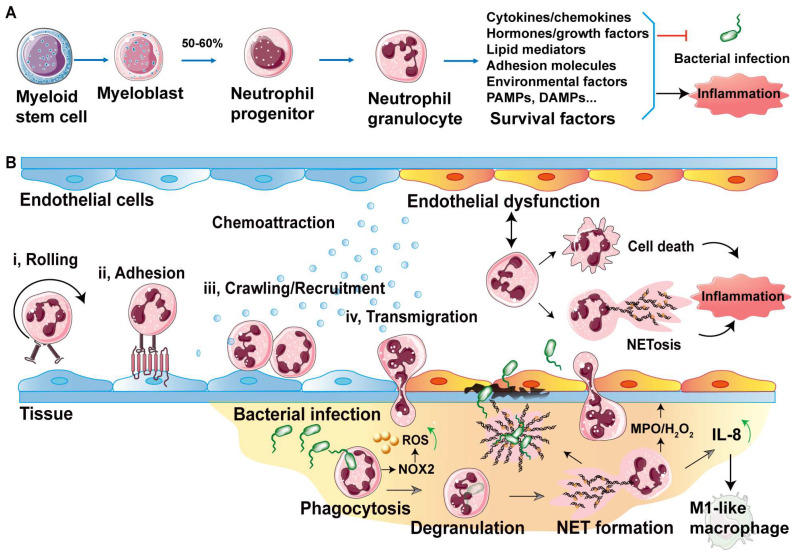
Overview of the interactions between the neutrophil and endothelial cells that combat invading bacteria. (**A**) Production of neutrophils, beginning from marrow and ending to meet bacteria. (**B**) Neutrophil-endothelial interactions show the defense steps against bacterial infection. Once sensing of a bacterial infection, rolling neutrophils (i) are immediately captured by endothelial cells by selectins, while adhesion (ii) is integrin-dependent. Crawling neutrophil (iii) arrives to perform recruitment (iii), inducing chemokine gradient and preparing for further transmigration (iv) across endothelial cells.

**Figure 2 biology-13-00374-f002:**
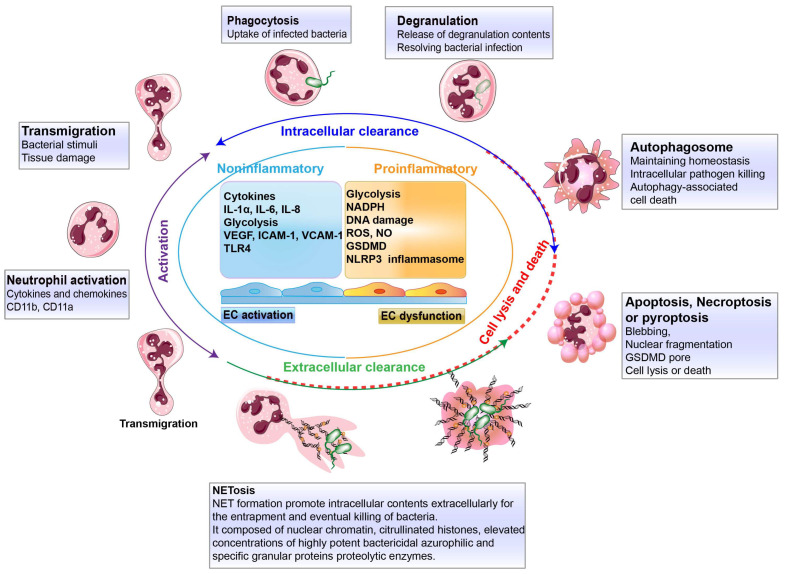
Neutrophil biological principles for defense against bacterial infection. Neutrophil activation and transmigration trigger two means of killing, including intracellular clearance (blue cycle line) and extracellular clearance (green cycle line). Intracellular clearance relieson neutrophil phagocytosis via the uptake of infected bacteria or the release of degranulation contents from intracellular components to kill bacteria. Autophagosome formation is also employed by neutrophils to clear intracellular bacteria. Extracellular clearance mainly depends on NETs. Meanwhile, the non-lytic form of neutrophil is NETosis, which is a cell death pathway. Also, neutrophil lysis and death (red cycle dotted line) can occur in apoptosis, pyroptosis, and autophagy-associated cell death.

**Figure 3 biology-13-00374-f003:**
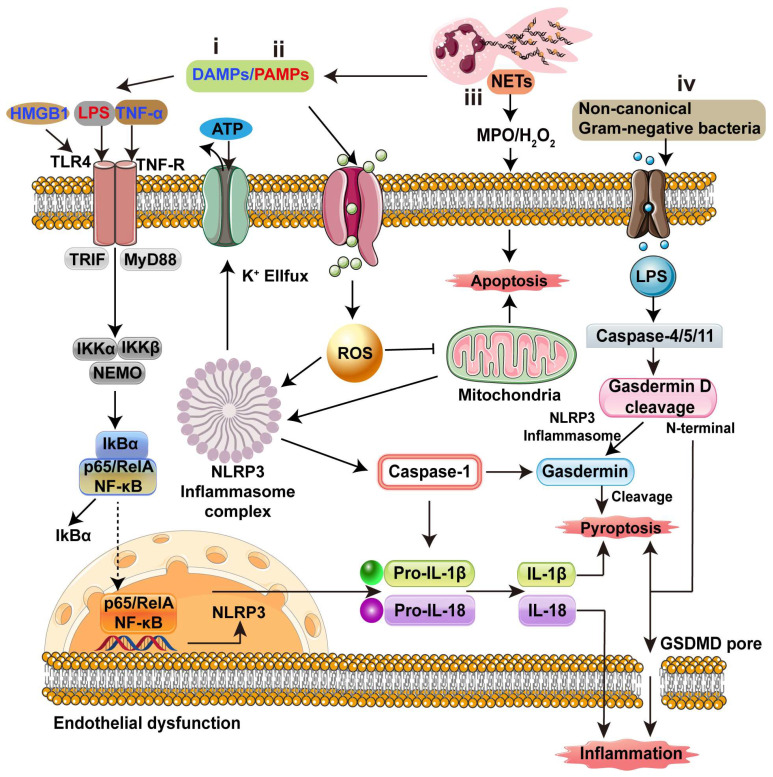
Inflammatory signaling cascades caused by endothelial dysfunction. (**i**,**ii**) PAMPs (e.g., LPS) and DAMPs (e.g., HMGB1; inflammatory cytokines like TNF-α or ILs) activate NF-κB signaling. PAMPs, DAMPs, and pore-forming toxins also induce ROS accumulation, which stimulates NLRP3 for IL-1β and IL-18 production, or Gasdermin for pyroptosis. (**iii**) NETs can also promote DAMPs and induce IL-1β and IL-18 production by activating NLRP3 through the LPS/TLR4/TRIF/NF-κB pathway and the TNF-α/TNF-R/MyD88/NF-κB pathway, enhancing endothelial cell inflammatory responses. (**iv**) Non-canonical Gram-negative bacteria induce intracellular lipopolysaccharide (LPS) accumulation, activating Caspase-4/5/11 and triggering the conversion of Gasdermin D into Gasdermin or GSDMD pore, leading to pyroptosis that enhances endothelial cell inflammatory response.

**Table 1 biology-13-00374-t001:** The sequential steps of neutrophil biology for tranendothelial migration against bacteria.

Basic Steps	Molecules		Reference
	Neutrophils	Endothelial Cells	
i, Rolling	PSG1, PTX3	P-selectin	[20]
	PSGL1, ESL1, CD44	E-selectin	[20,31]
	L-selectin	E-selectin	[20,23]
	CD95	CD95L	[6]
ii, Adhesion	β2 integrins LAF-1(α_L_β_2_)	ICAM-1	[7,20]
	Mac-1 (α_M_β_2_)	ICAM-2	[26,27]
	VLA-4	VCAM-1	[32]
	OSM	OSMR	[7]
iii, Crawling/Recruitment	β2 integrins Mac-1 (α_4_β_2_)	ICAM-1	[33,34,35]
	DPEP1 ligand	DPEP1	[36]
	GPR35	5-HIAA	[12]
iv, Migration	β2 integrins Mac-1 (α_4_β_2_), LFA-1	ICAM-1 and ICAM-2	[9,37]
	PECAM-1, JAM-A, CD99	PECAM-1, JAM-ABC, ESAM, VE-Cadherin, CD99, CD99L2	[11,38]

Note: PSGL-1, P-selectin ligand 1; PTX3, Pentraxin 3; ESL-1, E-selectin ligand 1; CD95L, CD95-ligand; VLA-4, very late antigen 4; ICAM, intracellular adhesion molecule-1; DPEP1, Dipeptidase 1; OSM, Oncostatin M; OSMR: OSM receptors; GPR35, G-protein coupled receptor; 5-HIAA, 5-hydroxyindoleacetic acid; PECAM-1, platelet endothelial cell adhesion molecule-1. JAM, junctional adhesion molecule; ESAM, endothelial cell-selective adhesion molecule.

**Table 2 biology-13-00374-t002:** The potential targets for inflammatory therapy by blocking endothelial cell dysfunction.

Potential Targets	Compounds	Mechanisms	Reference
Inhibit β_2_-integrin-mediated neutrophil recruitment	Soluble uric acid	Regulates intracellular pH and cytoskeletal dynamics	[75]
	hyperbaric oxygen	Reduces integrin-mediated adhesive properties of neutrophils	[76]
	Angiostatin	The kringles 1–3 and its kringle 4 directly interact with leukocyte beta1- and beta2-integrins	[77]
	herbimycin A	Inhibits of beta2 integrin-mediated tyrosine signaling	[78]
Inhibit NET formation	C34	Inhibits TLR4	[82]
	6-gingerol	Inhibits phosphodiesterase	[83]
	nNIF and NRPs	Blocks NET formation in vitro and in mouse models of infection and systemic inflammation	[84]
	DNase1	Preferentially cleaves protein-free DNA and degrades the scaffold of NET structures	[85]
ROS scavengers	Berberine	Reduces H/R-induced apoptosis and ROS production	[89]
	HSA	Reduces ROS and RNS production	[90]

## Data Availability

Not applicable.

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
