# Peer review of "Communications between Neutrophil–Endothelial Interaction in Immune Defense against Bacterial Infection"

_biology, 2024, doi:10.3390/biology13060374_

Round 1
Reviewer 1 Report
Comments and Suggestions for Authors
The manuscript covers an important topic, but could benefit from a more focused approach, clearer writing, and additional details and evidence to support the key points. Also, the manuscript effectively highlights the importance of the interplay between endothelial cells and neutrophils in the innate immune response against bacterial infections. The focus on understanding the mechanisms of transendothelial neutrophil killing and inflammatory cascades is well justified.
My comments:
1) The writing style is a bit unclear and could be improved for better readability. Some sentences are long and convoluted, making the overall flow and message less coherent. (e.g., the long, complex sentences in the first and second paragraphs in the Introduction Section).
2) The use of abbreviation could be better introduced and defined for readers who may not be familiar with them.
3) The language used throughout the manuscript could be improved to enhance clarity and concision. There are a few instances where the wording is awkward or convoluted, which can make the content less accessible to the reader.
4) The figures are unclear or low-resolution, work to improve them by using higher quality source images, adjusting the formatting and layout, or redrawing/recreating the figures as needed.
Overall, with some light editing and polishing, this manuscript seems ready for consideration as a short review article in the Biology journal.
Comments on the Quality of English Language
The language used throughout the manuscript could be improved to enhance clarity and concision. There are a few instances where the wording is awkward or convoluted, which can make the content less accessible to the reader.
Author Response
Journal: Biology
Manuscript: biology-2991045
Type: Article
Title: Communications between Neutrophil-Endothelial interaction in immune defence against bacterial infection
Correspondence: Prof. Xiaoye Liu
Authors: Zhigang Sun, Ruoyi Lv, Yanxin Zhao, Ziwen Cai, Xiaohui Si, Qian Zhang and Xiaoye Liu
Point-to-Point Response
Reviewer Comments:
Reviewer #1: The manuscript covers an important topic, but could benefit from a more focused approach, clearer writing, and additional details and evidence to support the key points. Also, the manuscript effectively highlights the importance of the interplay between endothelial cells and neutrophils in the innate immune response against bacterial infections. The focus on understanding the mechanisms of transendothelial neutrophil killing and inflammatory cascades is well justified.
Response: Thank you so much for your positive comments and constructive suggestions. We are glad that you find our topic is interesting and give us very useful suggestions. In the revised version, we have tried our best to revise the manuscript including in improving English written, revising Figures and Tables, suppling abbreviation section, simple summary and adding the examples of defense bacterial infection.
Comment 1: The writing style is a bit unclear and could be improved for better readability. Some sentences are long and convoluted, making the overall flow and message less coherent. (e.g., the long, complex sentences in the first and second paragraphs in the Introduction Section).
Response: Thank you for the insightful comments and suggestions. In the revised version, we had improved our English written throughout the manuscript. In addition, the clarified sentences were employed to completely replace the long sentences (See Introduction section 1, Page 2).
Comment 2: The use of abbreviation could be better introduced and defined for readers who may not be familiar with them.
Response: Thank you for your comment. We have supplied the abbreviation in the revised manuscript (See Abbreviation section, Pages 1-2).
Comment 3: The language used throughout the manuscript could be improved to enhance clarity and concision. There are a few instances where the wording is awkward or convoluted, which can make the content less accessible to the reader.
Response: Thank you for your detailed suggestions. We have replaced or revised the instances involving in the awkward or convoluted words. (See also the highlight words or sentences in the revised manuscript).
Comment 4:The figures are unclear or low-resolution, work to improve them by using higher quality source images, adjusting the formatting and layout, or redrawing/recreating the figures as needed.
Response: Thank you for your comment. We have already improved the images throughout the paper.
Comment 5: Overall, with some light editing and polishing, this manuscript seems ready for consideration as a short review article in the Biology journal.
Response: Thank you for your positive comments. In light of your suggestions, we have tried our best to revised this article. We believe that this revised version will be according with publication requirements.
Comment 6: Comments on the Quality of English Language. The language used throughout the manuscript could be improved to enhance clarity and concision. There are a few instances where the wording is awkward or convoluted, which can make the content less accessible to the reader.
Response: Thank you for your comment. The English language has been already revised and improved throughout manuscript.
Reviewer 2 Report
Comments and Suggestions for Authors
Manuscript entitled “Communications between Neutrophil-Endothelial interactions in innate defense against bacterial infection” require minor revisions, please check comments below
1. Introduction: a lot of statements with citations are used in introduction, please discuss your own observation, summarize in your own words
2. Please improve quality of figure 1 in manuscript with clear text
3. Line 73: “enter” not entry
4. Line 99- 112: Separate these sentences from figure 1, write separately, check all figures
5. Line 141-141: Revise this sentence
6. Line 146: no need to mention figure 2 here
7. Line: 150: “Meanwhile”
8. Line 152: cite reference
9. Line 159: mention specific bactericides released by neutrophils
10. Add separate headline/ paragraph about defense bacterial infection with examples, mention bacterial species
11. Improve quality of figure 2 in manuscript
12. Improve quality of all figures
13. Please summarize important points discussed in review in conclusion
Author Response
Journal: Biology
Manuscript: biology-2991045
Type: Article
Title: Communications between Neutrophil-Endothelial interaction in immune defence against bacterial infection
Correspondence: Prof. Xiaoye Liu
Authors: Zhigang Sun, Ruoyi Lv, Yanxin Zhao, Ziwen Cai, Xiaohui Si, Qian Zhang and Xiaoye Liu
Point-to-Point Response
Reviewer Comments:
Review 2#: Manuscript entitled “Communications between Neutrophil-Endothelial interactions in innate defense against bacterial infection” require minor revisions, please check comments below.
Response: Thank you for your comments. We have made the effort to respond to all comments and have revised the manuscript according to your suggestions. A point-by-point response is provided in following.
Comment 1: Introduction: a lot of statements with citations are used in introduction, please discuss your own observation, summarize in your own words.
Response: Thank you for your suggestions, we have added the observations and summarizes by our own words in introduction section of the revised manuscript. These words were highlight in yellow.
Comment 2: Please improve quality of figure 1 in manuscript with clear text.
Response: Thank you for your comment, we have resubmitted the high quality of Figure 1 as tiff version in the revised manuscript as also resubmit other figures. (See Figure 1, Page 4, line 132).
Comment 3: Line 73: “enter” not entry.
Response: Thank you for your detailed comment. We have revised it (See Pages 3, line 107).
Comment 4: Line 99- 112: Separate these sentences from figure 1, write separately, check all figures.
Response: Thank you for your comment. We have released the figure legend to main text.
Comment 5: Line 141-141: Revise this sentence
Response: Thank you for your comment, we have revised this sentence to “Neutrophils sense the signals of chemokines, cytokines and granular proteins de-rived from damaged endothelial cells to activate themselves for further fulfill the clearance mission”. (See Page 5, lines 167-169).
Comment 6: Line 146: no need to mention figure 2 here
Response: Thank you for your detailed comment, we have deleted the mention of figure 2.
Comment 7: Line: 150: “Meanwhile”
Response: Thank you for your detailed comment, we have deleted the repeated “Meanwhile”.
Comment 8: Line 152: cite reference
Response: We have cited the references 44 and 45
Comment 9: Line 159: mention specific bactericides released by neutrophils
Response: Thank you for your comment. We have added the specific bactericides released by neutrophils in the revised manuscript (See Pages 5-6, lines 196-200).
Comment 10: Add separate headline/ paragraph about defense bacterial infection with examples, mention bacterial species
Response: Thank you for your comment. In light of your suggestions, we first changed the headline of part 3 to “ Defence bacterial infection of Transendothelial Neutrophils” due to this part is originally supposed to discuss the role of neutrophils in fighting bacterial infections. Additionally, we included examples of bacterial species that are prevented by neutrophil defense.
Comment 11: Improve quality of figure 2 in manuscript
Response: Thank you for your comment. The quality of figure 2 has been improved in the revised version, as well as showed in below.
Comment 12: Improve quality of all figures
Response: Thank you for your comment, we have improved the quality of all figures in the revised manuscript.
Comment 13: Please summarize important points discussed in review in conclusion
Response: Thank you for your comment. The important points are discussed and summarized in conclusion in the revised manuscript.
Reviewer 3 Report
Comments and Suggestions for Authors![]()
In this manuscript, Sun et al. has provided a comprehensive review of the interactions between neutrophil and endothelial to defend against bacterial infection. The authors have summarized the main biological features of neutrophil and migration strategies. The molecular mechanisms and signaling pathways of intracellular and extracellular clearance by neutrophil have also been discussed. The inflammation related pathways caused by dysfunction of endothelial have been covered to study the potential therapeutic targets.
Specific comments:
1) Table 1, it’s better to have a figure to illustrate different steps in migration of neutrophil and the related biology mechanisms against bacteria.
2) The potential therapeutic targets and the related pathways and molecules can be summarized in a table.
3) The English language needs editing.
Comments on the Quality of English LanguageThe English language needs some editing.
Author Response
Journal: Biology
Manuscript: biology-2991045
Type: Article
Title: Communications between Neutrophil-Endothelial interaction in immune defence against bacterial infection
Correspondence: Prof. Xiaoye Liu
Authors: Zhigang Sun, Ruoyi Lv, Yanxin Zhao, Ziwen Cai, Xiaohui Si, Qian Zhang and Xiaoye Liu
Point-to-Point Response
Reviewer Comments:
Review 3#: In this manuscript, Sun et al. has provided a comprehensive review of the interactions between neutrophil and endothelial to defend against bacterial infection. The authors have summarized the main biological features of neutrophil and migration strategies. The molecular mechanisms and signaling pathways of intracellular and extracellular clearance by neutrophil have also been discussed. The inflammation related pathways caused by dysfunction of endothelial have been covered to study the potential therapeutic targets.
Response: Thank you for your valuable comments. We have taken the efforts to address all of your feedback and have incorporated your suggestions into the revised manuscript. A detailed response is provided below.
Comment 1: Table 1, it’s better to have a figure to illustrate different steps in migration of neutrophil and the related biology mechanisms against bacteria.
Response: Thank you for your comment. In fact, Figure 1B (step i to iv ) shows the different steps in migration of neutrophil. The correspondence step markers are supplied both in the Figure 1B and Table 1 in the revised manuscript. Figure 2 reveal the neutrophil biology mechanisms against bacteria.
Comment 2: The potential therapeutic targets and the related pathways and molecules can be summarized in a table.
Response: Thank you for your suggestion, we have already supplied the Table 2 as the potential therapeutic targets of inflammatory therapy.
Comment 3: The English language needs editing. Comments on the Quality of English Language. The English language needs some editing.
Response: Thank you for your comment. The English language have been improved in the revised version.
Round 2
Reviewer 1 Report
Comments and Suggestions for Authors
Dear Authors,
Thank you for your attention to my comments and effort on the current manuscript. After reviewing the revisions, I am happy to inform that the current form is suitable for publication after final confirmation by the editor-in-chief.
Best regards, AS